# Biodegradable Polylactide–Poly(3-Hydroxybutyrate) Compositions Obtained via Blending under Shear Deformations and Electrospinning: Characterization and Environmental Application

**DOI:** 10.3390/polym12051088

**Published:** 2020-05-10

**Authors:** Svetlana Rogovina, Lubov Zhorina, Andrey Gatin, Eduard Prut, Olga Kuznetsova, Anastasia Yakhina, Anatoliy Olkhov, Naum Samoylov, Maxim Grishin, Alexey Iordanskii, Alexandr Berlin

**Affiliations:** 1Semenov Institute of Chemical Physics, Russian Academy of Sciences, 119991 Moscow, Russia; 30111948l@bk.ru (L.Z.); akgatin@yandex.ru (A.G.); evprut@mail.ru (E.P.); 123zzz321@inbox.ru (O.K.); nastya_0496@mail.ru (A.Y.); aolkhov72@yandex.ru (A.O.); mvgrishin68@yandex.ru (M.G.); aljordan08@gmail.com (A.I.); berlin@chph.ras.ru (A.B.); 2Department of Chemistry and Physics, Plekhanov Russian University of Economics, 117997 Moscow, Russia; 3Department of Petrochemistry and Chemical Technology, Ufa State Petroleum Technological University, 450062 Ufa, Russia; naum.samoilow@yandex.ru

**Keywords:** polylactide, poly(3-hydroxybutyrate), poly(ethyleneglycol), blending under shear deformations, electrospinning, biodegradability, oil absorption

## Abstract

Compositions of polylactide (PLA) and poly(3-hydroxybutyrate) (PHB) thermoplastic polyesters originated from the nature raw have been obtained by blending under shear deformations and electrospinning methods in the form of films and nanofibers as well as unwoven nanofibrous materials, respectively. The degrees of crystallinity calculated on the base of melting enthalpies and thermal transition temperatures for glassy state, cold crystallization, and melting point for individual biopolymers and ternary polymer blends PLA-PHB- poly(ethyleneglycol) (PEG) have been evaluated. It has been shown that the mechanical properties of compositions depend on the presence of plasticizers PEG with different molar masses in interval of 400–1000. The experiments on the action of mold fungi on the films have shown that PHB is a fully biodegradable polymer unlike PLA, whereas the biodegradability of the obtained composites is determined by their composition. The sorption activity of PLA–PHB nanofibers and unwoven nanofibrous PLA–PHB composites relative to water and oil has been studied and the possibility of their use as absorbents in wastewater treatment from petroleum products has been demonstrated.

## 1. Introduction

In recent years the creation of biodegradable ecofriendly polymer materials capable to degradation under the environmental action forming substances safe for nature and thereby favoring the solution of ecological problems acquires a growing significance. These materials may be used in packaging as the biodegradable barriers and in responding to environmental challenging as the innovative pollution absorbents and separation membranes [1,2,3,4,5,6].

The biopolyesters, such as poly(α-hydroxyacides), namely polylactide (PLA), and poly(β-hydroxyacides) (PHA), namely poly(3-hydroxybutyrate) (PHB) as a basic homolog of the PHA family, are credible alternatives to the petrol-based plastics.

PLA is one of the most promising biopolymers obtaining on polymerization of D-, L- lactic acids originated through enzyme fermentation of renewable natural products such as corn, potato, sugar beet, etc. By a range of physical characteristics commercial-grade PLA resembles the biostable synthetic polymers polystyrene and poly(ethylene terephthalate) [7], but its biodegradability can occur only in hydrolytic and enzymatic media [8,9,10]. Being a good thermoplastic, it can be extruded into films and pellets, molded into goods of diverse shape and via electrospinning formed into ultrafine fibers [11,12]. Microbial PHB, which is more costly than PLA, is a fully biodegradable and renewable thermoplastic with the high crystallinity up to 70% brittle and high melting temperature. Owing to biocompatibility, biodegradability, and nontoxicity [13,14] PHB is used in different medical, packaging, and environmental applications, although its wide application is limited by current high cost. In fact, PLA, PHB, and their blends can be considered favorably for biomedical applications [15,16] especially in the tissue engineering, regenerative medicine and as the biodegradable therapeutic systems for drug delivery [17,18,19].

Both PLA and PHB application is limited by their mechanical properties, including brittleness, poor ductility, caused by high crystallinity, and relatively low thermostability due to a small difference between temperatures of melting and decomposition. According to the previous comprehensive studies [20,21,22,23], blending of these two biopolyesters, being quite similar in chemical structure, is very promising for elaboration of novel ecocompatible materials with improved transport and mechanical characteristics. For example, in the work [24] the authors showed that PHB incorporation into PLA films strongly increased oxygen barrier and simultaneously decreased wettability that is quite reasonable for food packaging application. Moreover, blending of PLA and PHB has considerable promise, since the addition of PHB to PLA endows the compositions with biodegradability, and PLA allows one to reduce the cost of the corresponding products as compared with the cost of PHB based materials [25]. The studied compositions can be prepared by different methods determining their structure, properties, and hence the application areas. The innovation methods for obtaining of these compositions such as blending under shear deformations and electrospinning are believed to be of great prospect.

It is noteworthy, that combining PLA and PHB as the binary blend without a structure-modifying additive has rarely led to miscibility of the polymer components. To resolve the above drawbacks including their rigidity some of authors have produced PHB-PLA compositions by using plasticizers such as limonene [26], trybutyrine [27], oligomeric lactic acid [28], polyethylene glycol (PEG) [24], and others. Due to strict requirements to food packaging demands, a series of conventional plasticizers such as t-butylphenyl phosphate and dioctyl phthalate should be excluded. In this context, the application of a series of low molar mass PEGs as the nontoxic plasticizing agents open the promising prospects for the more effective design of PLA-PHB systems. Given the PEG benefits as an ecofriendly and nontoxic plasticizer, it should be noted that its effect on thermal, mechanical and diffusion characteristics of the triple system have been explored in very few cases. According to Scopus^®^ search engine data, the corresponding results of low molar mass PEG in the region of 400–1000 impact have not been published yet besides some rare papers, where PEG was used as the additive with a single molar mass, e.g., 300 [29].

The blending of PLA with PHB is often carried out in solution (regularly in chloroform) that requires essential volumes of solvent and the process becomes durable and costly. However, the blending under high-intensity shear deformations in the absent of solvents [30] allows one to obtain PLA–PHB compositions by ecologically safe procedure and in this case their properties are determined by the mixture composition.

At the same time electrospinning presents a promising method for obtaining fibers in the micro- and nano-range. Nanosized fibrous structures formed during electrospinning are characteristic of high specific surface area and controlled porosity that makes promising their application as efficient filters, high-sensitivity sensors, and absorbents with a high sorption capacity [31,32]. Recently, Liu, Cao, Iordanskii et al. [33] have reported the production and characterization of PHA-PHB spun fibers produced by melt electrospinning. In the following paper of the same authors [34], the comparison of melt electrospun PHB-PLA fibers and molded films with analogous compositions has been presented to elucidate the diffusivity impact on drug release features. In addition to the usage of electrospun fibers in biomedicine as well as at development of innovative packaging materials, they are starting to be actively used for the development of innovative absorbents for pollutants removing [35]. Therefore, a current tendency to production of ecologically safe high-selectivity nanofibrillar porous sorbents on the basis of PLA–PHB compositions obtained by electrospinning for recovery of oil pollutants from wastewater is of great practical interest.

Considering the arguments presented above, this work is devoted to preparation of PLA-PHB compositions by two different methods: via blending with low-molecular plasticizer PEG under shear deformations and electrospinning as well as subsequent examination of their reasonable characteristics. This study involves primarily the food packaging elaboration and ecofriendly fibrillar absorbent application for oil pollutions removing.

## 2. Materials and Methods

### 2.1. Materials

PLA (the trade mark “Ingeo™ 4043D”) is a product from Nature Works (Minnetonka, MN, USA) as pellets with a diameter of 3 mm was used for blending with PHB and PEG. According to the manufacturer, the weight average (Mw) and number average (Mn) molar masses of PLA are found to be 2.20 × 10^5^ g/mol and 1.65 × 10^5^ g/mol respectively with a polydispersity index Đ = Mw/Mn = 1.35. Preliminary the pellets were dried in a vacuum oven for 8 h at 70 °C. The PLA density was 1.245 g/cm^3^ and transparency 2.1%. This trade mark of PLA designated for food packaging was proved in the list of FDA effective [36].

PHB with weight average molar mass (M_w_) being equal to 2.05 × 10^5^ was kindly provided by Biomer company (Krailling, Germany), personally by Dr. U. Haenggi. The biopolymer was originally in the form of a white powder with particle size of 3–7 µm. It has density of 1.290 g/cm^3^; 0, 43 mas % humidity, and therefore the material was dried overnight at 70 °C before mixing. In 2007 the FDA had cleared its marketing in the USA, indicating a bright future for a practical application of PHB in biomedical areas, e.g., as medical devices [37].

Poly(ethylene glycol) (PEG), CAS Number: 25322-68-3, with M_w_ equals to 400, 600, and 1000 g/mol was supplied by Sigma-Aldrich S.A. PEG has been found to be nontoxic and is approved by the FDA for use as excipients or as a carrier in different pharmaceutical formulations, foods, and cosmetics.

Chloroform, CAS Number 67-66-3, was also purchased from Sigma-Aldrich (St. Louis, MO, USA). 

All materials were used as received, without any further purification. 

### 2.2. Production of Compositions

The polymer compositions were prepared under action of shear deformations in a Brabender type mixer (Plastograph Brabender^®^ GmbH and Co. KG, Duisburg, Germany) at 180 °C for 10 min at the rate of 100 rpm. For further investigation, films of 0.3 mm thick were pressed on a Carver press (Carver Inc., Wabach, IN, USA) at 180 °C and pressure of 10 MPa for 10 min followed by cooling with the rate of 15 °C/min. As a result, the films of 0.2–0.25 mm thick were formed.

### 2.3. Investigation of Thermophysical Properties

Thermophysical characteristics and thermal stability of the initial polymers and their blends were studied by the differential scanning calorimetry (DSC) method on a DSC-204 F1 calorimeter (Erich Netzch GmbH and Co. Holding KG, Selb, Germany) at the heating rate of 10 K/min in the temperature range of 30–200 °C. Sample weight was approximately 10 mg. In the air, the experiment involved the first heating, cooling, and the second heating with the same rate. 

### 2.4. Mechanical Tests

Mechanical characteristics of samples were determined on an Instron-3365 tensile machine (High Wycombe, UK) under uniaxial stretching, at the rate of the upper traverse motion of 50 mm/min at room temperature. From the obtained diagrams, elastic modulus *E*, tensile strength *σ*_p_, and elongation at break *ε*_p_ were calculated. The results were averaged by five samples.

### 2.5. Biodegradability

The biodegradation was estimated by carrying out microbiological tests of the compositions with different component ratios for stability to the effect of mold fungi. The principle of the method is in the holding of samples infected with an aqueous suspension of mold fungi spores under the optimum conditions of their growth and the estimation of fungi resistance (the index of fungi growth) of samples by variations in their color, shape, and texture according to the six-point scale. 

The biodegradability of compositions was studied also by modeling the processes occurring under the natural conditions. For this purpose, the samples were placed into a container with wet soil (pH = 7.5) meant for place growing. The containers were kept in a thermostat at 30 °C during several months. The rate of biodegradation in soil was evaluated by the weight loss of samples at regular time intervals.

### 2.6. Atomic Force Microscopy (AFM)

The fiber surfaces were studied on a Solver HV (NT-MDT) atomic force microscope. The tests were carried out in the semicontact mode with the use of a HA_NC probe sensor (chip size 3.6 mm × 1.6 mm × 0.45 mm, cantilever size 87 µm × 32 µm × 1.75 µm, curvature radius of a tip 10 nm, force constant 5.8 N/m).

### 2.7. Production of Fibers and Unwoven Materials by Electrospinning 

The 7% solution for electrospinning was prepared by mixing for all PLA/PHB ratios (1/0, 0.5/0.5, 0.1/0.9, and 0/1) in cosolvent, chloroform, at magnetic stirring for 8 h at room temperature. The electrical conductivity of polymer blend solutions, as one of key factors being responsible for fiber perfection, were 10 µS/cm. After mixing the solutions were placed in a 20 mL syringe fitted with a stainless-steel needle with inner diameter of 0.1 mm. The fibers were produced using the laboratory installation described earlier [31] as a single-capillary vertical setup with the following operative parameters such as flow rate of 0.5 mL/h, voltage of 12 kV, and the distance between the needle tip and the electrode collector was 18 cm.

### 2.8. Oil Absorption Measurements 

As sorbate was used, West Siberian crude oil with a density of 848 kg/m^3^ (according to RF standard: GOST 51069-97) and a sulfur content of 10 mas. % (according to standard RF: GOST 32139-2013) was also used. Oil absorption properties for electrospun fibrils forming the corresponding mats with different PLA–PHB compositions (100:0, 50:50, 10:90, and 0:100 wt %) were evaluated by a weight method with the use of an HR-200 analytical balance (A&D Weighing Co, Milpitas, CA, USA) with a readability of 10^−4^ g. 

### 2.9. Statistics and Data Availability Statement

The mat thicknesses were equal to 86 ± 4.1 μm with accuracy 0.0434 for all the polymer ratios. The averaged diameter of electrospun fibers was calculated by seven independent measurements of AFM microphotographs and was depended on the PLA–PHB ratio in the range from 8.3 ± 0.9 to 11.9 ± 1.4 µm. The measurement error of absorbed oil after triple weighing was ± 4.6%. Temperature was measured to an accuracy of 0.1 °C. Relative standard deviation of mechanical testing for *E,*
*σ_p,_* and *ε*_p_ with 5-fold averaged values was spanned in a 5–10% interval. 

## 3. Results and Discussion 

### 3.1. Thermophysical Properties of PHB, PLA, and their Compositions

PLA–PHB blends with different component ratios were obtained in a Brabender mixer under conditions of shear deformation. Since PLA and PHB are rigid and brittle polymers, their blending was carried out in the presence of PEG of various molar mass (400, 600, and 1000). For further thermophysical and mechanical measurements as well as the biodegradability evaluation, films from PLA, PHB, and their blends were formed.

Figure 1 shows DSC curves of PLA (*1*,*1′*), PHB (*2*,*2′*), and their blend with PEG (PLA–PHB (80: 20 wt %) + 5 wt % PEG_400_) (*3*,*3′*) at the first (without prime) and second (primed) heating. It follows from the figure that the DSC curve of the first heating displays four peaks belonging correspondingly to glass transition (T_g_ = 65 °C), cold crystallization (T_cc_ = 124 °C), and melting doublet (T_m1_ = 159.4 °C, T_m2_ = 161.8 °C). At the second heating, because of change in PLA crystalline structure, the temperatures of transitions decreased on several degrees (Table 1).

As it is shown in Figure 1, the DSC curve for high crystalline PHB demonstrated a single melting peak, and the peak reflecting glass transition was absent due to minor content of amorphous phase. The determined PHB crystallinity at the first and second heating was 49% and 52% correspondingly. For the neat PHB after the first heating the T_m_ slight reduction was also associated with crystalline structure reorganization. In the blends, owing to a sharp decrease in PHB concentration by 4 times, the evaluation of its crystallinity was practically impossible therefore corresponding values are not given in Table 1.

In Figure 2 the DSC thermograms for the ternary blends are presented. In DSC curves of PLA–PHB blend with 5% PEG_400_, at the first heating, five peaks were observed due to glass transition, cold crystallization, melting temperature doublet for PLA, and melting temperature of PHB (Table 1). Depression of PLA transition temperatures, T_g_ and T_cc_, in the blends as compared to the analogous temperature transitions of the neat PLA was due to the presence of two other components, PHB and especially PEG, which affect the mobility of PLA molecules. At the second heating of the triple blends, all temperatures of the thermal effects (melting and cold crystallization) remain the same while the glass transition peaks of PLA were absent due to the crystallinity increase (Table 1).

Exploring the PEG impact on thermal blend behavior, it is possible to find that the general patterns of the DSC curves changed. In particularly, the PEG_400_ content increment up to 10% led to vanishing the PLA glassy transition and simultaneously to increasing to maximal crystallinity of 57%. By acting as the plasticizer, PEG declined the positions of T_g_ and T_cc_ on the temperature scale. Moreover, the most pronounced effect of plasticization, expressed as the low temperature shift, the growth of crystallinity, and the decrease of T_cc_ values was found for the blends containing PEG_400_ that is the plasticizer with the lowest molecular weight.

Generally, the glass transition temperatures (T_g_) of the blends indicate polymer miscibility degree. If blending leads to the composition having a lower single T_g_ than the brittle component, namely PLA, that implies the emergence of flexible chains, thereby lowering brittleness and, hence, improving applicability. Otherwise, if the binary composition has a higher single T_g_, the brittleness could arise because of macromolecular stiffness increasing [40]. Two or even more glass transition points could testify immiscibility of the blend [41] especially when both polymer components are crystallizable [42]. In our case, the DSC curves and data in Table 1 show the single T_g_ that is more typical for glassy transition of PLA. From the same data, it follows that T_g_ in all ternary blends containing 5% PEG was essentially decreased. This decrease was especially remarkably observed for PEG of the lowest M_w_ (400) that testified the plasticizing impact of PEG in the range of M_w_ from 400 to 1000.

It is well established that the low molecular PEG was characterized by low molar volume and high concentration of terminal hydroxide groups. These characteristics promoted its diffusivity into the biopolymer blends and facilitated the interaction with the functional groups of the biopolyesters (PHB and PLA) that could lead to the plasticizing effect [43,44]. In contrast, high molecular PEG is denied such an opportunity and in that case the plasticization does not occur at all [45]. For example, recent research [46] has shown that relatively high molecular PEG (M_w_ > 2000) was not only a poor plasticizer, but its loading has caused well-observed phase separation as well.

A pronounced effect of PEG addition was observed for PLA-PHB blends as T_g_ depression for all plasticized samples in the informative works [24,46]. Following the concept of the work [24], we reckon that in the framework of free volume theory [47,48], the plasticizing effect of PEG increases the segmental mobility of the biopolymers and, ultimately, enhancing molecular contact between PLA and PHB molecules. For all the ternary blends PLA-PHB-PEG, the shift of T_cc_ values to the field of lower temperatures is evidence of PEG plasticizing effect and PLA segmental mobility enhancing. This effect can be demonstrated more clearly if instead of T_cc_ values, the difference between two characteristic temperatures, namely T_cc_ – T_g_ = ∆T_ccg_, should be used. The results of the calculation are presented in Table 1.

The feature ∆T_ccg_ reflects the temperature interval between the glassy-rubber transition and the development of cold crystallization, e.g., the situation when the segmental motion becomes high enough and a low temperature crystallization as the dynamic process can happen. It should be noted that the narrower this interval, the more effective the PEG action as the plasticizer. With this consideration, in contrast to PEG_1000_, PEG_400_ along with PEG_600_ could be treated as the most efficient plasticizer. Enhancing in PLA segmental mobility as the expression of plasticizing effect is confirmed also by the doubled growth of PLA crystallinity degree in the ternary blends (Table 1), namely from 27% to 42% and to 55 % for 5% and 10% of PEG_400_ content correspondingly.

### 3.2. Mechanical Testing

By the data of mechanical testing for PLA, PHB, and their blends containing PEG of different molar masses, elastic modulus *E*, tensile strength *σ_b_,* and elongation at break *ε_b_* of the initial polymers and their blends were evaluated. The mechanical parameters of PLA and PHB are characteristic of glassy polymers with a low breaking strain. Characterization of PLA with *E* = 2700 MPa, *σ_b_* = 45.4 MPa, and *ε_b_* = 4% demonstrated its high rigidity, while PHB is an even more rigid polymer with higher elastic modulus and lower elongation at break, such as *E* = 2900 MPa, *σ_b_* = 19.5 MPa, and *ε_b_* = 1%. In the literature the brittleness and the poor ductility of both PLA and PHB have been broadly discussed to avoid the basic drawbacks that prevent their widespread applications, see e.g., [49,50,51]. As mentioned above in the Introduction, simple blending of these two biopolymers has rarely led to their mechanical improvement. On the contrary, their characteristics even have worsened as it was reported in the paper [52], where *σ**_p_* and *ε**_p_* values of PLA–PHB blends decreased especially at a higher PHB content in the blends.

Figure 3 shows the dependences of *E*, *σ_b_*, and *ε_b_* values for PLA–PHB–PEG compositions (5 wt % PEG) on the PHB content in blend. As can be seen from the data in Figure 3a, an increase in the content of more rigid polymer PHB in the ternary compositions practically does not influence on elastic modulus of the blends. However, as it follows from the curve in Figure 3b, the gain in the PHB content leads to some decrease of *σ_b_* value monotonically and the lowest value of *σ_b_* observed for a composition involving 30 wt % PHB. Additionally, finally, an increase of the PHB content in compositions resulted also in a drop of elongation at break (Figure 3c). The observed scatter in the experimental values was supposedly connected with structural inhomogeneity of PLA-PHB compositions.

Figure 4 shows dependences of mechanical parameters of PLA–PHB–PEG_400_ compositions on the PEG content (the PLA–PHB ratio is equal to 80:20 wt % for all compositions). As can be seen from Figure 4a,b elastic modulus and tensile strength of compositions decreased as the PEG content in compositions grew. Elongation at break for the above mentioned blends at small PEG contents remains very low (from 1% to 2%) and an increase of its value begins with further gain in the plasticizer content (Figure 4c). For compositions with 10% PEG_400_, supposedly the character of deformation changes from brittle fracture to plastic flow; as a result, elongation at break attains 13%, elastic modulus decreases to 800 MPa, and tensile strength decreases from 28.2 to 11. 6 MPa. So, the elastic modulus was dropped approximately three times, tensile strength decreased by 2.5 times, and elongation at break increased by more than seven times reaching the value of 14%. Hence, as in the case of thermal findings (Section 3.1), the ternary composition revealed itself as essentially more flexible than binary PHB–PLA system.

For PLA-PHB compositions the loading of different plasticizers is a quite common way to improve both mechanical and thermal behavior of these binary blends. In the paper [53] Averous and Martin have considered the influence of low molecular PEG and some other plasticizers on thermal and mechanical characteristics of PLA. For the PLA-PEG system they have shown the remarkable decrease of T_g,_ and as in the case of our findings, PEG_400_ showed the higher thermal effect comparing to PEG_1000_. Simultaneously under storage and loss modules measurement the presence of 20% PEG_400_ had demonstrated the better effect comparing to other plasticizers. The ternary PLA-PHB-PEG blends remain scantily investigated, except of two relevant publications, namely of the paper of Wang et al. [54] considering the similar composition PLA-PHBHV-PEG and the work of Arietta and coworkers [29] that has described the analogous ternary blend PLA-PHB-PEG. In the first of the mentioned publications the PEG addition led to an essential improvement of tensile and impact characteristics of PLA-PHB blends. Particularly, impact strength of the ternary composition was 2–4 times higher than that of the initial binary PLA-PHBV blend, as well as the elongation at break.

It should be pointed out, that the change of M_w_ of PEG in the range 400–1000 practically did not affect the mechanical characteristics of the ternary system (Table 2). However and in this case the slight decline in the basic characteristics, namely E и σ_b_, was observed in the presence of the PEG_400_, which had the lowest value of M_w_.

### 3.3. Biodegradation of Composites 

In the previous paper the authors investigated the hydrolysis of PLA, PHB, and PHB-PLA blends in phosphate buffer [55]. In this work, Bonartsev et al. showed that the rate of hydrolytic degradation of PLA-PHB blend differed from the corresponding homopolymers. Most recently, in the paper of Bonartsev, Berlin, Iordanskii et al. [56] it was demonstrated the impact of the PHB+HV (hydroxyvalereat) copolymer content on the hydrolytic behavior of the membranes. Additionally to the results obtained in aquatic medium, it would be interesting to explore the PLA-PHB compositions under fungi contact as the reasonable situation during contact the films with a microbial medium.

As the capacity of decomposition under the environmental action is one of the most important properties of the biodegradable composites the biodegradation of PLA, PHB, and their composites was investigated by two independent methods: by the estimation of the degradation under the action of the set of fungi as the biodegradation agents and by the exposure in soil. The tests on the fungus resistance performed with PLA, PHB, and their blends with PEG showed the difference in the intensity of mold fungi growth.

The data on the fungi resistance of samples depending on the testing time with determination of the fungi species are given in Table 3.

As can be seen from the data given in Table 3, PLA is virtually not subject to degradation under the fungi action, whereas, with PHB, the maximum intensity of mold fungi growth (5 points) was found. In PLA–PHB–PEG composition (70: 30 wt % + 5 wt % PEG_1000_), an insignificant growth of mycelium was revealed in tenth day of testing, in this case as well as for the initial PHB, the spore formation was provided by *Aspergillus brasiliensis, Trichoderma virens,* and *Paecilomyces variotii* species. A decline in PHB concentration to15 wt % is accompanied by the decrease of fungi growth intensity from 4 to 2 points.

The photographs of PLA–PHB–PEG compositions before (a) and after (b) exposure in soil during 3 months are present in Figure 5. As is seen from the data obtained, after exposure in soil the films demonstrated spots, which are the first stages of the biodegradation process.

These data coincided with our previous works, where by using the SEM method it was shown that on the compositions PLA-cellulose [57], as well as the on compositions PLA-starch [58] after exposure in soil the deep cracks and the open-end holes on the surface of the films are formed. These cracks and holes are the initial stages of the sample fragmentation leading to their following destruction due to the biodegradation.

The estimating the capacity of the studied compositions of biodegradation on exposure in soil the weight loss was measured at intervals. It occurs that the weight loss of the investigated composite film after exposure in soil within 12 months was approximately 10%.

In the same time, for a pure PLA film, during 12 months exposure in soil weight loss was absent, whereas, for PHB, the weight loss was equal to 36% within 6 months, and further exposure in soil resulted in the complete disintegration of sample film.

Thus, the study of fungi resistance of PLA, PHB, and their compositions showed that unlike PLA PHB was a fully biodegradable polymer. These data coincided also with the results on weight loss obtained at the exposure of these polymers in soil. Generally, the biodegradability of the investigated PLA-PHB films depends on the composition and increases with the PHB content.

### 3.4. Oil Sorption by Electrospun Fibrous Absorbents

For estimating the potential application of PLA/PHB blends particularly for oil recovery from aqueous media, the unwoven fibrous materials in the shape of plane fibrillar mats were prepared by solution electrospinning. As a result, the 2D structures were obtained with the developed interfibrillar pores system and the high specific surface. The water permeability and the absorption selectivity of the above membranes relative to organic components make them promising products used for separation of oil–water systems.

The AFM study of the fiber surfaces demonstrated the distinction in their morphology that could clearly be observed only at the large magnification (Figure 6). Whereas the PLA-PHB blend fibers (1:1) had a relatively uniform surface without the noticeable roughness, the homopolymers PHB and PLA demonstrate surface heterogeneity. The cause of heterogeneity may be determined by the rapid solvent evaporation and the crystallinity of the polyesters, in which the size of crystals (spherulites and lamellae) is comparable with the fiber diameter.

The sorption capacity of unwoven fibrous materials (mats) of PLA, PHB, and their compositions with different component ratio (Table 4) was determined by measuring the sorption of oil from aqueous medium at the oil layer thickness no less than 10 mm. The homogeneous polyesters PLA and PHB proved to be characteristic of high oil absorption (30 and 45 *g/g*, respectively), whereas their blends demonstrated a decrease in this fiber characterization namely 16 and 15 *g/g* for 50: 50 and 90: 10 wt % respectively. Thus, the mechanism of oil absorption by the unwoven materials under consideration is a relevant example of oil sorption on the surface of oleophilic fibers as well as the formation of films between fibers through surface tension forces [59].

As was mentioned above, the biodegradability of PHB significantly differed from PLA biodegradability, besides, its cost exceeded the cost of PLA about twice. To decrease the PHB-PLA blend absorbent expenses and to vary the rate of their biodegradation, it is advisable to use these electrospun fibers for oil recovery via absorption and following utilization in soil. Absorption capacity comparison for several promising absorbents designated for oil recovery from aqueous media shows that the maximal productivity demonstrated exfoliated graphite (EGS). From the Table 4 follows, PHB and PLA fibers were somewhat inferior to it and their blend fibers had absorbed this pollutant even less. However, the high hydrophobicity of the biopolyesters’ mats, its continually falling prices due to cheaper raw materials for their large-scale production, and being natural renewable resources, as well as their ecofriendly character of biodegradation creates for them a promising prospect for making novel materials being able to replace a nonbiodegradable absorbent.

## 4. Conclusions

PLA-PHB compositions were prepared by environmentally safe methods, namely by blending under shear deformations in the absent of solvents and electrospinning of biodegradable blends. Preliminary literature analysis partly presented in the article has shown that PLA-PHB compositions were both semicrystalline polymers with poor miscibility. This circumstance encouraged the authors to explore thermal and mechanical characteristics of PLA-PHB blends modified by low molecular PEG as the plasticizer. The crystallinity and the specific thermal transitions (T_g_, T_cc_, and T_m_) for the individual PLA, PHB, and their compositions with PEG were calculated. Thermal and mechanical features of the ternary blends testified the pronounced behavior of PEG as the plasticizer with the noticeable increase in the elastic modulus, tensile strength, and elongation at break and with the remarkable decrease of T_g_ that was particularly evident for the PEG_400_ with the lowest Mw. It was shown that the second heating resulted in a positive increment of the crystallinity degree demonstrating a significant importance of the thermal prehistory of the blend samples.

On estimation of the fungi resistance of polyesters and their compositions, it was found that, in contrast to PLA, PHB was a completely biodegradable polymer, and the biodegradability of related compositions depended on the component ratio and increased with the PHB content. 

Unwoven fibrous materials on the basis of electrospun fibers of PLA, PHB, and PLA-PHB compositions with different component ratios were characteristic of a high absorption capacity relative to oil and moderate water absorption that allowed one to consider these materials as efficient absorbents of oil products on ecological accidents and water pollution. The biodegradability of the developed fibrillar systems makes possible their utilization after service completion presented also their important advantage to successfully solve environmental challenges.

## Figures and Tables

**Figure 1 polymers-12-01088-f001:**
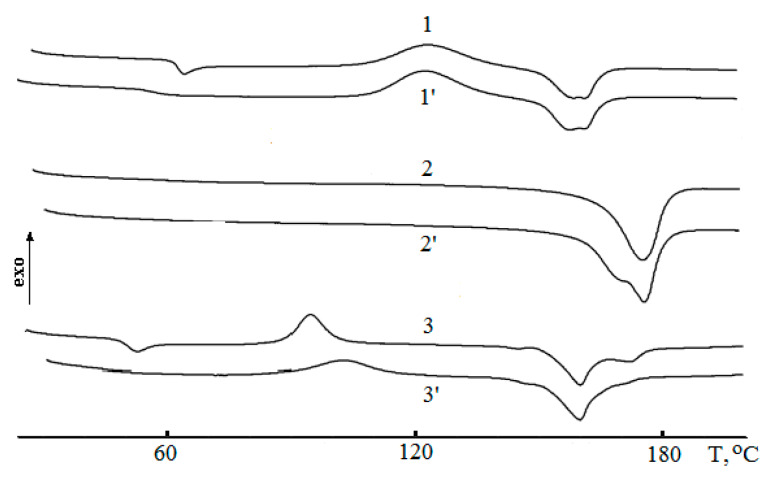
Differential scanning calorimetry (DSC) curves of PLA (*1*,*1′*), PHB (*2*,*2′*), and their blend with PEG (PLA–PHB (80: 20 wt %) + 5 wt % PEG_400_) (*3*,*3′*) at the first (without prime) and second (primed) heating.

**Figure 2 polymers-12-01088-f002:**
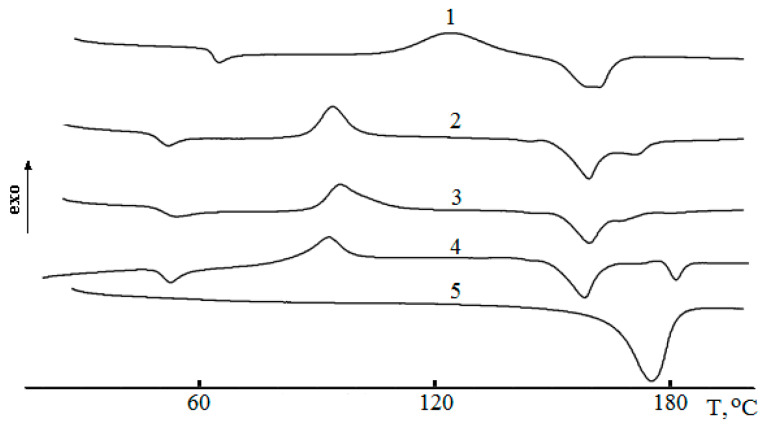
DSC curves of PLA (1), PHB (5) and PLA–PHB blends (80: 20 wt %) with 5 wt % PEG of different molar mass 400 (2), 600 (3), and 1000 (4).

**Figure 3 polymers-12-01088-f003:**
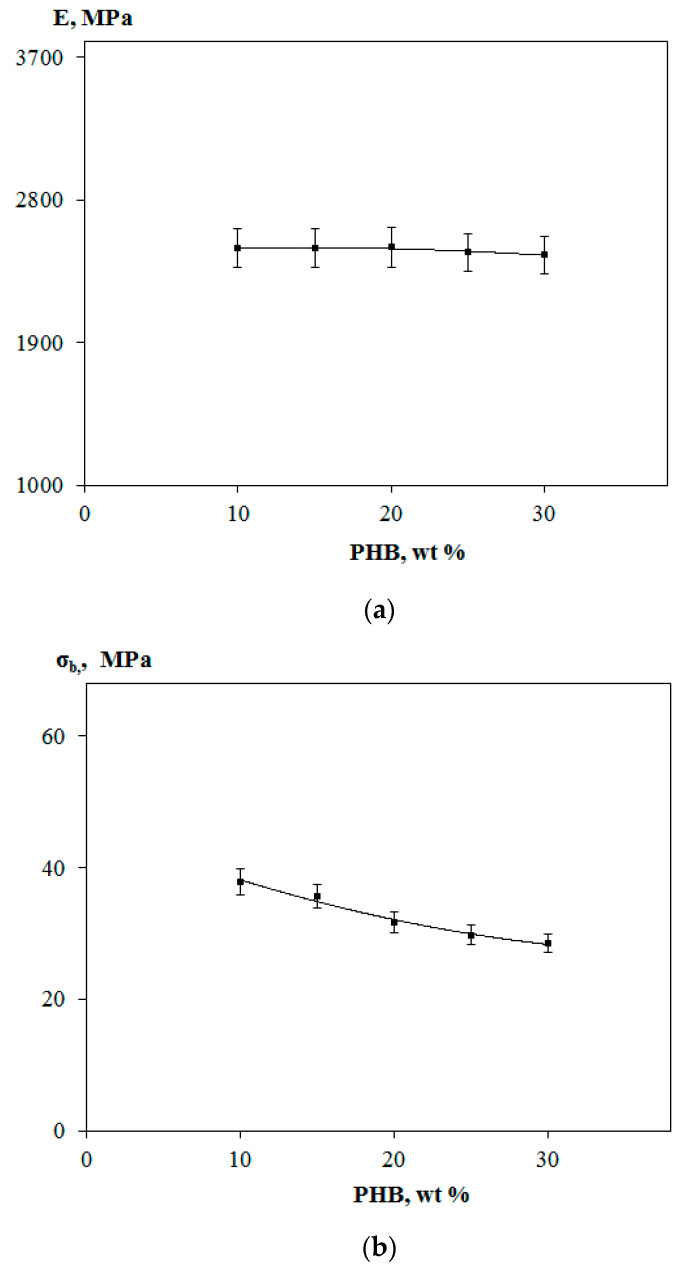
Elastic modulus *E* (**a**), tensile strength *σ_b_* (**b**), and elongation at break *ε_b_* (**c**) of PLA–PHB–PEG_1000_ compositions vs. PHB content.

**Figure 4 polymers-12-01088-f004:**
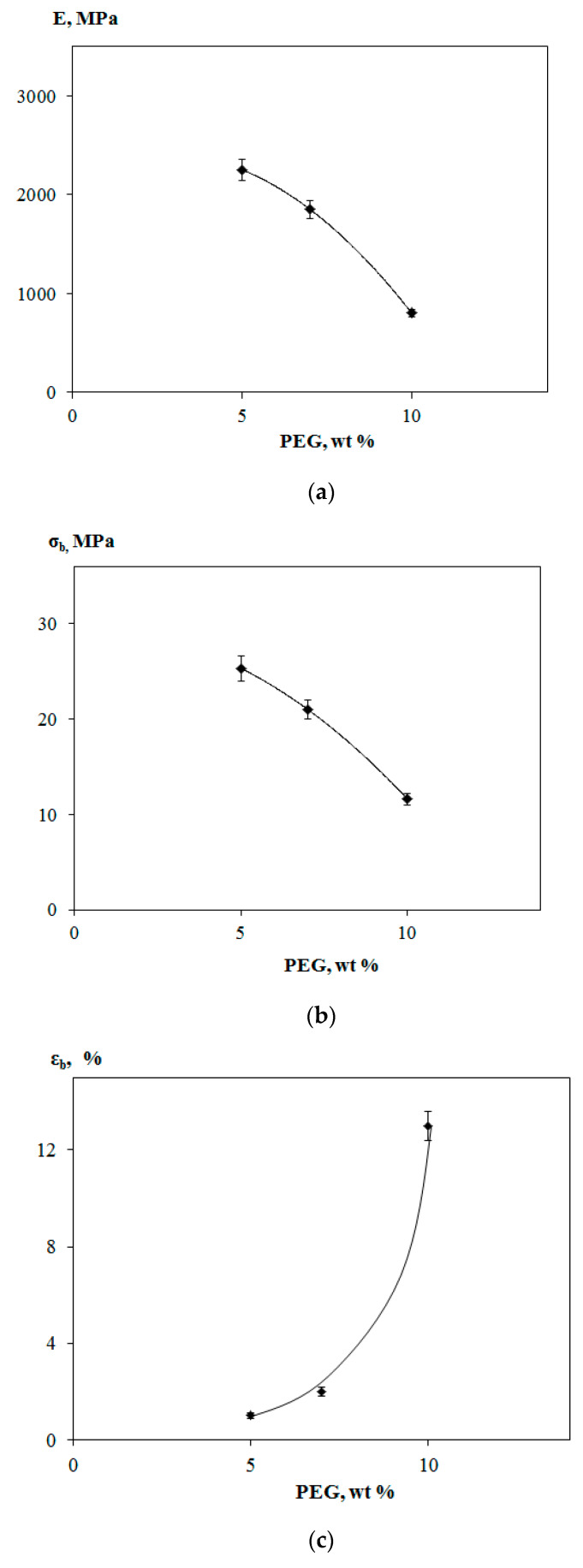
Elastic modulus *E* (**a**), tensile strength *σ_b_* (**b**), and elongation at break *ε_b_* (**c**) of PLA–PHB–PEG_400_ compositions vs. PEG content.

**Figure 5 polymers-12-01088-f005:**
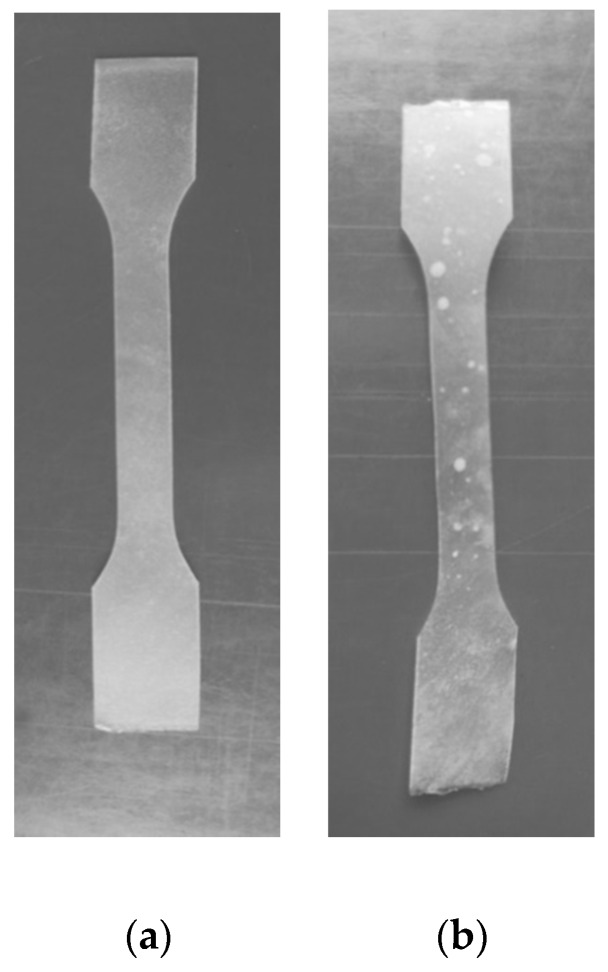
Photographs of PLA–PHB composition films (80: 20 wt % + 10 wt % PEG_400_) before (**a**) and after (**b**) exposure in soil within 3 months.

**Figure 6 polymers-12-01088-f006:**
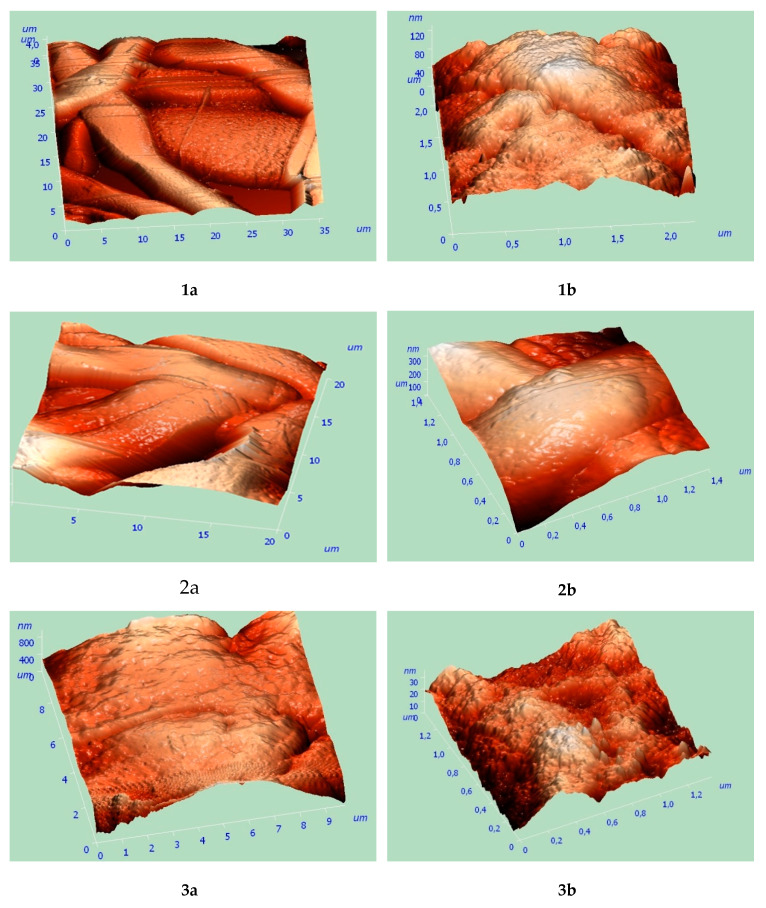
3D Atomic Force Microscopy AMF images of fibers surfaces for PLA (1), (2) PLA–PHB (50: 50 wt %), and (3) PHB fibers at different magnification (**a**,**b**).

**Table 1 polymers-12-01088-t001:** Thermophysical characteristics of initial PLA and in its blends of various compositions at the first and second heating.

Blend Composition PLA-PGB, wt %	Heating	T_g_, °C	T_m_, °C	ΔH_m_, J/g	T_cc_, °C	∆T_ccg_, °C	χ_cr,_% *
Polylactide							
PLA = 100	first	65.0	doublet 159.4; 162.0	25.6	124.0	59.0	27.3
	second	63.0	doublet 158.4; 161.7	27.7	123.3	60.3	29.6
(80:20) + 5% PEG_400_	first	52.0	doublet 144.4; 159.3	31.3	94.0	42.0	42.0
	second	−	doublet 146.4; 159.3	36.6	102.0		49.0
(80:20) + 10% PEG_400_	first	−	157.6	41.3	83.0		55.0
	second	−	156.8	42.7	doublet 81.5; 127.0		57.0
(80:20) + 5% PEG_600_	first	54.2	doublet 145.6; 159.3	22.8	96.0	42.0	30.0
	second	−	doublet 148.0; 159.4	32.4	106.0		43.0
(80:20) +5% PEG_1000_	first	53.5	159.0	29.6	107.0	53.5	34.0
PHB = 100	first	-	175.0	71.0	-		49.0
	second	-	170.0	75.5	-		52.0

* Crystallinity of individual PLA and PHB was calculated by formula $\chi_{cr}=\frac{\Delta H_{\exp}}{\Delta H_{m}^{o}}$, where *ΔH_m_^0^* are melting enthalpies of PLA and PHB at 100% crystallinity equal to 93.7 [38] and 146 J/g [39], respectively. Crystallinity of PLA in blends *χ_cr_* was calculated by formula $\chi_{cr}=\frac{\Delta H_{\exp}}{\Delta H_{m}^{o}\cdot W_{PLA}}$, where *W_PLA_* is weight fraction of PLA in blend.

**Table 2 polymers-12-01088-t002:** Influence of PEG with different molar mass on mechanical properties of PLA-PHB blends.

Blend Composition, PLA-PHB, wt %	E, MPa	σ_b_, MPa	ε_b_, %
80:20 + 5% PEG_1000_	2500 ± 125	31.6 ± 1.6	2 ± 0.1
80:20 + 5% PEG_600_	2700 ± 130	21.0 ± 1.1	1 ± 0.1
80:20 + 5% PEG_400_	2250 ± 110	25.3 ± 1.2	2 ± 0.1

**Table 3 polymers-12-01088-t003:** Fungi resistance of samples in points depending on testing time.

Testing Time, Days	Fungi Growth Intensity, Points			
	PLA	PHB	PLA-PHB (70:30 wt %) + 5 wt % PEG_1000_	PLA-PHB (85:15 wt %) + 5 wt % PEG_1000_
10	0	2	1	1
15	0	2	1	1
21	0	2	2	1
28	0	5	2	1
50	0	5	3	1-2
84	0-1	5	3-4	2

**Table 4 polymers-12-01088-t004:** Oil absorption for the productive polymer absorbents.

Absorbent	Oil Capacity, *g/g*	Absorbent Condition	References
PDOS	3.4	Peat Dust Oil Sorbent	[60]
Lessorb^TM^	5.6	Dispersed powder	[60]
PET	14	Nonwoven Fabric	[61]
PHB-PLA (90:10)	15	Nonwoven mat	[61]
PHB-PLA (50:50)	16	Nonwoven mat	[61]
Sintapeks	24	Cotton-processing product	[62]
PLA	30	Nonwoven mat	[61]
PU	37	Foam pellets (5-8 mm)	[61]
PHB	45	Nonwoven mat	[61]
EGS	50	Exfoliated Graphite Sorbent	[60]

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
