# Peer review of "Biodegradable Polylactide–Poly(3-Hydroxybutyrate) Compositions Obtained via Blending under Shear Deformations and Electrospinning: Characterization and Environmental Application"

_polymers, 2020, doi:10.3390/polym12051088_

Round 1

Reviewer 1 Report

The manuscript may be approved for publication, however, it need some improvemets. Below are presented remarks which should be taken into account before publication.

Abbreviation MM for the molar mass  should not be used as there are generally accepted abbreviations for molar mass depending on the type of average molar mass is discussed (Mn, Mw, etc.). Moreover MM is not defined. Molar masses as well as Đ in Materials part should be presented to both: PLA and PHB. These data are important from scientific point of view. What average molar mass of PHB is referred, Mn or Mw? It is recommended by IUPAC to use “molar mass” rather than molecular mass or molecular weight so please use such term.

The authors use the terms molecular mass  and molecular weight (molecular weight is acceptable) interchangeably but it seems that "molar mass" due to IUPAC regulations will be the best way to standardize the text.

The sentence starting on line 215 is a bit risky. It is the effect of comparing only two lines in Table 1. The differences between the compared values are insignificant and, additionally, there is no data regarding the effect of PEG1000 (there are data related to PEG 1000 but it is difficult to compare them with the previous ones since the composition of the blend is 90/10; to compare the influence of certain value it is necessary assume the others as immutable, e.g., composition 80/20). The same applies to the sentence in lines 234-235. In table 2 there are data concerning the PEG100 and blend composition 80/20, it means you have them, so simply put them also into table 1 but in form presented in this Table.

Line 403: The authors state with oil absorption for PLA / PHB 50/50 is 15.7%, and for 10/90 15.2 g / g, however, the figure shows other values. Additionally, there are still doubts as to why to use PLA mixtures with relatively expensive PHB, and with much lower absorption properties is a good solution. It is uncertain whether biodegradability will be a particularly important property of electrospun mats containing absorbed oil.

66 tributyrine jest trybutyrine

74 is proved should be approved

Reviewer 2 Report

accept

Author Response

Dear reviewer, please accept our thanks for revised our manuscript.

Reviewer 3 Report

In the response to reviewer document, the authors of this work have stated a long page to show that their electrospinning process of PLA/PHB fibers is correct and the resulted 12um diameter fibers are nanofibers (see Line 14 in Page 1). And then, they also stated that they have published lots work in electrospinning PLA and PHB.

Thus, I made some work, and found that they already published the wholeelectrospinning section elsewhere. One paper has been published in a Russian journal entitled "New fibrilar composites on the base of biodegradable polyethers poly (3-hydroxybutyrate) and polylactide with high selective absorbtion of oil from water medium" with doi of 10.31857/S0869-56524875528-531, and another paper has been published in an international journal entitled "New Fibrillar Composites Based on Biodegradable Poly(3-hydroxybutyrate) and Polylactide Polyesters with High Selective Absorption of Oil from Water Medium" with doi of 10.1134/S0012501619080049. All these two paper have same sample compositions, AFM images and oil absorption ratios with this manuscript.

Thus, I have a strong ethical concern about the entire work. And I am afraid I have to refuse this review or to reject this work.

Round 2

Reviewer 1 Report

The manuscript should be read thoroughly again by the author to avoid typing errors, e.g. in Table 1 in the headings is PLA-PGB ect.

Reviewer 3 Report

accept

This manuscript is a resubmission of an earlier submission. The following is a list of the peer review reports and author responses from that submission.

Round 1

Reviewer 1 Report

In this paper, the authors claimed that they fabricated PLA/PHB composites through both melt blending and electrospinning, and studied their properties. However, in my opinion, this cannot be called a paper, it is more like a document that consists several experiment reports. In this work, the authors used a mixer to fabricate PLA/PHB with three different PEG, and studied the DSC. But no scientific discussions have been made. Moreover, in the mechanical properties section, they compared the tensile properties of PLA/PHB/PEG1000 with different PHB loading. Then, they provided tensile properties of PLA/PHB/PEG400 with different PEG400 loading. I cannot see any interconnections and  reasons for these descriptions. If you want to discuss the effect of PHB and PEG on the blends’ mechanical properties, you should keep the same type of PEG you used. Moreover, the figures in this work are hard to see. Besides, in the second part of this work, the authors electrospun PLA with PHB, but the fabrication details are missing in this report i.e. the concentration of polymers. Besides, as measured in AFM, such ~10 um diameter fibers cannot be called ultrafine fibers. What’s more, it is not easy to produce such large fibers through electrospinning, the authors should carefully check their electrospinning setup and methods. After all, the reviewer has no judgement on this work.

Reviewer 2 Report

The submitted manuscript described a kind of composite material PLA-PHB with better biodegradation ability, and its adsorption properties to water and oil were studied. The reviewer believes the authors need to perform several extra experiments and revise the manuscript before it can be accepted for publication. Here is the detail of necessary revision, 1. The DSC image of PHB was included in Figure 2. Please supplement thermophysical characteristics of PHB at the first and second heating in Table 1. 2. When studying the influence of PHB and PEG content on the mechanical properties of blends, the authors needed to consider the influence of PEG with different molecular weight. 3. Results and discussions section, the authors needs to analyze the results rather than simply describe the measured data (such as sections 3.2 and 3.3). 4. Add error bars to the Figure 3 and 4. The source of oil-water mixture for adsorption need to be supplemented by the authors. 5. Please cite more updated references in results and discussions.

Reviewer 3 Report

The manuscript is interesting, however, several issues should be considered before being accepted.

  1. lines 39-43: PLA is much more widely used in regenerative medicine and the pharmaceutical industry. So maybe write they both have such uses.In addition, when it comes to degradation, PLA degrades in water or pH 7.4 buffer at a much faster rate than PHB.
  2. Characterization of melt electrospun fibers and films based on PLA-PHB blends uwere described by the Authors in Macromol. Symp. 2018, 381, 1800130 and this work should be mentioned.
  3. The aim of the work should be clearly stated at the end of the Introduction.
  4. Ref 12 concerns the PEG/starch blends. Is it well cited?
  5. In recent years, a lot of attention has been paid to PLA/PHB blends, also plasticized. Authors during the discussion of results should refer to these results, e.g. J Polym Environ DOI 10.1007/s10924-014-0654, although PEG 300K was used in this work.
  6. Authors should explain/justify why they study biodegradation only using fungi. Moreover there are also presented results of materials biodegradation in soil which is slow. The soil should be defined similarly to the test conditions (see Biomacromolecules 2006, 7, 3125-3131). Sample weight loss during experiment was observed, which may suggest that biodegradation of the sample occurs, however, it is highly likely that during the test hydrolytic degradation also occurs, which would be visible through a decrease in the molar mass of the components.
  7. Examination of sorption properties proves that PLA and PHB have a much better oil sorption capacity than their mixtures. Admittedly, both PLA and PHB are relatively expensive materials, so what is the reason for using their mixtures, which are characterized by much worse oil sorption properties.
  8. Please check carefully the manuscript to avoid the Russian letters (see line 118).